# About the Importance of Planning the Location of Recycling Stations in the Urban Context

Mats Wilhelmsson 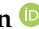

Department of Real Estate and Construction Management, Division of Real Estate Economics and Finance, Royal Institute of Technology (KTH), SE-100 44 Stockholm, Sweden; matswil@kth.se

**Abstract:** Recycling is essential to the circular economy and reduces the environmental impact of our consumption. Creating conditions for recycling in new residential areas is relatively easy but finding good recycling opportunities in existing residential areas is more complicated. The recycling of newspapers, plastic and glass must be relatively close to where people live; at the same time, the locations must be relatively discreet and not disturb the residents in the area. The purpose of the article is to analyse the effect of small and local recycling stations (RCSs) on the attractiveness of residential areas. This has been made possible by analysing housing values for almost 200,000 housing units near 250 RCSs in Stockholm, Sweden. Using an identification strategy that relies on postal code fixed effects, we find evidence that the proximity to RCS affects housing prices on average in both owner-occupied single-family houses and cooperative owner-occupied apartments (condominiums). The results indicate that proximity to the RCS is negatively capitalised in housing values (the effect amounts to approximately 1.3 percent of the housing values), which indicates that the city should consider this in its planning.

**Keywords:** recycling; housing values; capitalisation; circular economy; Stockholm; Sweden





## 1. Introduction

If net-zero emission by 2045 is to be achieved, more raw materials will need to be recycled more efficiently. Producer responsibility for recycling packaging has existed in Sweden since the early 1990s, and the climate benefits of recycling are significant. Calculations indicate, for example, that for every kilogram of recycled plastic, carbon dioxide emissions are reduced by up to two kilograms [1]. Even when considering the resources used for collection, transport, and sorting, the reduction in $CO_2$ is still nearly 1.5 kg. According to the Swedish Environmental Protection Agency, Sweden must have no net emissions of greenhouse gases into the atmosphere by 2045 to have negative emissions. Since emissions per person amount to nine tonnes per year, recycling is only one measure among many needed to reduce the total climate impact. However, material recycling is an integral part of the overall transformation.

A circular economy with efficient recycling permeates all consumption and production; a society based on such an economy will achieve high sustainability goals. Recycling is of utmost importance for the circular economy through increased recycling in production and consumption (see [2–4]). Recycling materials such as glass, paper, and plastic are essential for a more circular economy and crucial for preventing material from being deposited or thrown away in nature. For example, recycling plastic is vital to reducing microplastics in nature and water [5]. A good overview of the literature on the relationship between packaging, sustainability, and circular economy is a recent paper by Sastre et al. (2022) [6].

Recycling stations (RCSs) must be accessible to increase households' recycling. Recycling opportunities must be close to households, and as long as it is not possible in existing residential areas to cost-effectively create the opportunity to recycle in the property, RCSs must be placed locally in the residential areas [7–9]. According to Sidique et al. (2010) [10],

recycling behaviour is affected by the distance to RCSs. The closer the distance is between the recycling station and the dwelling, the more frequently the recycling station will be used. Miller et al. (2016) [11] analyse the effect of accessibility to recycling stations and its visibility on how much is not recycled. As a case study, they use recycling in university buildings. Their results indicate that neither availability nor visibility significantly impacts recycling. The results, however, indicate that the combination of accessibility and visibility is important. Li et al. (2020) [12] show similar results. DiGiacomo et al.'s research in 2017, on the other hand, shows that accessibility and easy access to recycling has a major impact on the behaviour of households in dysfunctional family homes [13]. They conclude that their findings should influence waste management and environmental policy.

The design of the recycling stations themselves is not the only factor to have a potential negative impact on how much is recycled. Keramitsoglou and Tsagaraki's (2018) [14] research shows the potential of involving the residents in designing the recycling stations as it both increases the acceptance of the recycling stations in the living environment, and potentially increases the recycling. The importance of the different elements of design is also something that Jiang et al. (2021) [15] point out in their research.

Several studies analyse the effect of waste disposal by incineration, such as a recent article by Zhao et al. (2016) [16] and an older study by Kiel and McClain (1995) [17]. However, those facilities are much larger than RCSs and expected to significantly negatively affect housing values. For example, [16] found that property prices fell by as much as 25 percent near an incineration plant, and [17] found that this devaluation starts even before the plant is in place and that the effect is persistent for several years after the plant is in operation. Eshet et al. (2007) [18] analyse the effect on the property market near waste transfer stations using the hedonic methodology, or capitalisation effect. Their explanation for why they can see this negative externality is that the waste transfer stations bring disamenities such as noise, odour, litter, vermin, visual intrusion and perceived discomfort. The effect is apparent, and the closer the dwelling is to the waste transfer stations, the more negatively the housing value is affected. However, the size of these waste plants is significantly more extensive and has a much more economically significant impact on the surrounding environment than the RCSs analysed here.

Increasing the number of RCSs and placing them close to residential areas has already occurred in Sweden. There are approximately 5000 RCSs in Sweden and as many as 250 RCSs in a city of a million inhabitants, such as Stockholm. The location of the stations is vital for increasing households' recycling rate, but they are also potentially an externality in the urban environment. RCSs entail traffic, noise, dirt, and a potential health risk. The location is thus essential not only relative to the recycling volume but also the attractiveness of the residential areas. However, few studies have been conducted on the negative externality of RCSs.

This study aims to contribute knowledge about the impact of RCSs on the attractiveness of residential areas. Recycling is important and will become increasingly more critical in the future to create a sustainable society; nevertheless, as they are designed and located today, RCSs may have local costs that could and should be avoided to gain greater acceptance and higher welfare gains.

The location of an RCS is important for creating opportunities for the efficient collection and recycling of packaging materials. Local installation of RCSs must be cost-effective for emptying and cleaning, both in number and location. At the same time, the RCS must create as little nuisance as possible to residential attractiveness. It is essential for housing satisfaction, yet to our knowledge, no studies have analysed the possible effect of RCS location on housing attractiveness. Hence, we aim to provide a significant contribution in producing a basis for planning where these RCSs should be located to minimise the socioeconomic cost.

An important issue in these analyses of capitalisation effects is the question of endogeneity. Have RCSs been located in areas with low residential attractiveness, or has the location of the RCS created areas with lower residential attractiveness? One of our contri-

butions is that we systematically address this issue. In our analysis, we have addressed this problem by using a few excluded yet relevant explanatory variables, namely fixed effects, a treatment effect model, a propensity score method, and a micro-analysis of the immediate area around each recycling station.

Section 2 will present the theoretical model and methodological approach. Decisive for interpreting the relationship between housing prices and RCSs is that we do not have problems with endogeneity, which will permeate the chosen method. Section 3 presents the selected case study focusing on locating RCSs in Stockholm. This is followed by Section 4, where data is presented and described. The empirical economic analysis is presented next in Section 5, and the article closes with a conclusion and policy-relevant questions in Section 6.

## 2. The Theoretical and Methodological Framework

The research aims to increase knowledge about the impact of recycling centres on attractiveness of locality. We do this by analysing the prices of privately owned homes. The theoretical starting point is Rosen (1974) [19], where the housing price is a function of its attributes. In the first step, a hedonic price equation is estimated, where estimated parameters can be interpreted as marginal willingness to pay for each attribute. The estimated parameters can thus be interpreted as implicit prices (hedonic prices). The attributes consist of the characteristics of the apartment or property and the residential area. The methodology is often used to estimate implicit prices for different types of negative externalities, such as traffic noise [20], or positive externalities, such as shopping malls [21]. Here we will test the hypothesis that proximity to RCS negatively capitalises on housing prices by including the distance to the recycling station in the hedonic price equation; and, as an alternative, including it as a binary variable to indicate whether the dwelling is within a specific range of the recycling station. The hedonic price equation that will be estimated looks like Equation (1),

$$HP_{i,t} = \alpha_j + \beta_1 X_{i,t} + \beta_2 RCS_{i,t} + \beta_3 T_t + \varepsilon_{i,t} \tag{1}$$

where *HP* is equal to house prices (all models are estimated with a price as a natural logarithm based on a Box-Cox transformation), and the matrix *X* represents all value-affecting attributes such as size, age, and location. The variable *RCS* represents proximity to a recycling station. In the empirical analysis, proximity to an RCS is used as a binary variable or as the shortest distance to an RCS. Our hypothesis is that $\beta_2$ is negative, and the vector *T* is a binary variable measuring the month the dwelling was sold (fixed time effects). The subscripts *i* and *t* indicate transaction and time. All Greek letters indicate parameters that are estimated. The parameter $\alpha$ has a subscript of *j* for a postal code, indicating that fixed urban effects are included in the model.

What can be expected when it comes to the correlation of proximity to RCS and housing prices? A large part of the negative impact depends on where and in what context the RCSs are located. If they are located on a minor street close to parks and green areas, the recycling station may be considered polluting the environment; but if it is located on a major road with much traffic and near a gas station or adjacent to a shopping centre, it can be expected that the effect is significantly less or negligible. The management of the RCS is also essential. How often they are emptied and if it is often messy has an expected adverse effect. For example, Mattsson Petersen and Berg (2004) [22] asked some individuals about the importance of managing their RCS. The majority stated that they thought the care was good or better, but the variation was large between them. At one station, as many as 48 percent of respondents stated that the cleanness of the area around the RCSs was bad or very bad.

Another factor to consider may be how much traffic is generated to and from RCSs. It may also be reasonable to expect that the capitalisation effect regarding an RCS will vary depending on whether it is near single-family houses compared to condominiums. The survey results in [22] indicate that visitors to the RCS are on their way to other activities,

i.e., the visit is not the trip's primary purpose. Furthermore, they observed that 90 percent of the visits took place by car, which indicates that the location of the RCS is not only important in terms of logistical and cost-effectiveness, but that the location itself generates traffic that can be disruptive. Car traffic has a negative impact on housing prices [20], but the increased traffic generated by the RCS can be marginal, depending on the existing traffic volume. Minor streets in single-family housing areas can significantly increase traffic volume due to the recycling station, while if the street is already a major road, the traffic from and to the recycling station is relatively insignificant.

Underlying factors that positively affect price are the size of the dwelling, measured as the total square meters of living space, and the number of rooms. We analyse both the owner-occupied condominium housing market and the single-family housing market. In the condominium case, the fee to the tenant-owner association has a negative price effect. There is also an expected price premium for houses closer to the CBD [23,24], and the same applies to the proximity to public transportation [24,25] and shopping centres [21].

The independent attributes must be exogenously given to interpret estimated parameters as implicit prices and thus marginal willingness to pay. In the presence of endogeneity, estimated relationships are just relationships, not causal ones. It is usually no problem to assume that the independent variables are exogenously given for all apartment and property attributes, but this is more difficult for many residential area attributes. Perhaps among these is the attribute of primary interest in this study, namely proximity to RCS. There are several reasons why this attribute might be endogenous, including reverse causality, omitted variables, and measurement errors [26–28].

### 2.1. Reverse Causality

One reason for endogeneity is that RCSs are located in low-priced locations rather than in the surrounding areas. For example, RCSs may be located near major roads, petrol stations or similar places. The estimated relationship between housing prices and proximity to RCS will not, in those cases, be causal, and it can even denote reverse causality.

The methodology employed assigns certain transactions as treatment and compares these with other transactions that are assigned to the control group, a quasi-experimental design. It is a treatment effect model similar to Heckman's treatment effect model without instrument variables as in [26,28], but the group is not randomised. Hence, there is undoubtedly a treatment selection bias. We have tried to mitigate this effect by using a propensity score methodology [29], where the observations in the treatment area are as similar as possible to those in the control group in all respects other than proximity to the RCS. The methodology has been used in previous analyses such as [30,31]. The optimal way to handle the problem would have been to use the difference-in-difference, instrument variable or regression discontinuity design methodology. Unfortunately, no data on transactions before the current location of the RCSs is available, which renders these methods inaccessible.

The treatment effect model is a two-step model where, in the first step, we define treatment and control groups and calculate the probability that the observation is included in the treatment group. In the second step, a weighted least square model will be estimated where the probability is the inverse of the weights. In this way, we analyse whether the observations in the treatment group are as similar as possible to the observations in the control group. The method is suggested by [32] and used in, e.g., [30]. The propensity score equation looks like Equation (2),

$$P(Z) = \Pr(Treat = 1 | Z) \tag{2}$$

where the propensity score (*PS*) is the probability of treatment (*Treat*) given the covariates *Z*, with $0 \leq P(Z) \leq 1$, and the weighted hedonic least square model looks like Equation (3),

$$HP_{i,t} = \alpha_j + \beta_1 \frac{X}{PS_{i,t}} + \beta_2 \frac{RCS}{PS_{i,t}} + \beta_3 Treat \frac{1}{PS_t} + \varepsilon_{i,t} \tag{3}$$

where *PS* is the estimated propensity score. The higher the probability that the dwelling is similar to the properties close to the RCS, the greater the observation's weight in the estimate.

Moreover, we have visually inspected all RCSs to ensure no justification for reverse causality between housing prices and proximity to the RCS. The inspection has also allowed us to classify the RCSs based on characteristics in the geographical location. This, too, has been done to minimise the potential endogeneity problem. All RCS locations have been classified as good or bad locations. The hedonic price equation that has been estimated looks like Equation (4).

$$HP_{i,t} = \alpha_j + \beta_1 \frac{X}{PS}_{i,t} + \beta_2 \frac{RCS\ (good)}{PS}_{i,t} + \beta_3 \frac{RCS\ (bad)}{PS}_{i,t} + \beta_4 Treat \frac{1}{PS}_t + \varepsilon_{i,t} \quad (4)$$

We hypothesise that $\beta_2$ and $\beta_3$ are negative and that $|\beta_2| > |\beta_3|$.

### 2.2. Omitted Variables

The empirical analysis will not use panel or pure cross-sectional data but instead pooled cross-sectional data. Fixed effects for time and fixed effects for residential areas have also been implemented by using postal code information. Our goal is to reduce the problem of omitted variables and thus the endogeneity problem by including fixed effects. They will be effective if the spatial effect is constant within the group or invariant over time [26]. There are a large number of hedonic studies that address spatial heterogeneity by including fixed effects, such as [33–35]; nevertheless, as the authors of [36] point out, a large number of included fixed effects can result in few degrees of freedom, which impairs the model's accuracy.

For this reason, spatial fixed effects are used to control spatial dependency and omitted variables. However, fixed property effects have also been included in the analysis of condominiums, as condominiums in the same properties have the exact same coordinates and information about repeated sales in the single-family housing data. This is another possibility to minimise the problem of omitted variables and thus the potential endogeneity problem. The inclusion will also mitigate the effect of spatial dependency to some degree, as the fixed property effect aims to check for heterogeneity at the property level. The methodology will be effective if properties at the property level are constant over time, although this may be too questionable an assumption. Another weakness is that the inclusion of the fixed property effect dramatically reduces the degree of freedom and makes the estimation significantly more computationally challenging. If the number of analysed observations is large, the problem of many fixed effects is not severe.

### 2.3. Measurement Error

Of course, it can also be the case that the endogeneity problem is due to measurement errors in the variables examined, notably in the RCS variable. In order to eliminate problems with measurement errors regarding RCSs, each location, according to the FTI's register, has been checked in Google Map and Google Street View to coordinate correctly the location of the RCS. Street addresses tend to place the property's coordinates some distance from the roadside where the RCSs are often located. Thus, the endogeneity problem caused by measurement errors in the location of the RCS is minimised.

### 2.4. Robustness Test

As a robustness test, we have also (1) changed the assumptions about the treatment and control area, respectively, and (2) randomised where the RCSs are located. The latter is done by "moving" RCSs 0–500 m from their original location. After that, the hedonic price equation was estimated again, with the assumption that RCSs should not have a negative capitalisation on housing values. The test is similar to the placebo test commonly used in the regression discontinuity design methodology, see e.g., [37–39] in a difference-in-difference context.

In summary, endogeneity is controlled by including fixed area variables. In this case, these included postal codes, visual inspection of the vicinity around the RCS, and any additional area attributes. In a sub-analysis, property fixed effects are used after restricting the area to a smaller inner-city location where RCSs are more commonly found on smaller streets in residential areas.

## 3. Recycling and the Case of Stockholm, Sweden

### 3.1. Recycling in Sweden

In Sweden, there are more than 5000 RCSs. In 2020, the annual collection result per inhabitant in Sweden was just over 22 kg of glass, 17 kg of packaging paper (cardboard), almost 9 kg of plastic, just under 2 kg of metal, and 14 kg of newspapers. An essential prerequisite for this accomplishment is that RCSs in the built environment are accessible and make recycling easy. Sweden has had a far-reaching producer responsibility since 1994 regarding the collection and reuse of packaging materials, and the RCSs we analyse here are an essential part of this responsibility.

Recycling behaviour across municipalities in Sweden varies. Hage and Söderholm's (2007) [40] results indicate that the variation between municipalities regarding collection can be explained by differences in demographic and socioeconomic factors, environmental preferences, geographical differences, and local policies. The authors of [41] examined RCSs in a small town in Sweden at the end of the last century. The purpose was to create a basis for planning future locations of RCSs by examining attitudes toward collection and how much was collected at each station, i.e., the volume of recycled material. Both are important questions to determine where and how many RCSs the city should plan for.

There is an ongoing discussion begun in 2021/22 about what the system of RCSs should look like in the future. One proposal discusses how recycling can be made closer to the property by collecting it in the property or its vicinity. The proposal is out for consultation, and the government is expected to decide about the future system in June 2022.

### 3.2. Recycling in Stockholm

Our case study is the city of Stockholm, the capital of Sweden. As of 2020, it has nearly 1 million inhabitants. Stockholm is divided into 13 different districts. By population, Södermalm is the largest district, followed by Hägersten-Älvsjö and Enskede-Årsta-Vantör. In the year 2020, Spånga-Tensta and Skärholmen were the smallest districts, with around one-third of the population of Södermalm.

In the city of Stockholm, there are 250 RCSs. This system is an integral part of recycling paper, glass, plastic, metal packaging, and batteries. Research results from [22] showed that the most common visitor to recycling stations brought paper, newsprint, and glass packaging; the least common items were batteries and textiles. Of course, this may have changed since the survey was conducted.

FTI (Förpackning och Tidningsinsamlingen) owns and operates these RCSs. Together with the municipality, they decide where they should be placed, what should be collected, and how often they should be emptied. The ownership of FTI consists of four material companies and was formed in connection with the government's decision on producer responsibility for packaging in 1994. Table 1 shows the number of RCSs in different parts of Stockholm per 100,000 inhabitants and 1000 hectares of land.

The number of RCSs varies between the different districts in Stockholm, from only 6 to as many as 35. Naturally, this is mainly due to the number of residents in the district. More densely populated neighbourhoods also have more RCSs, but that is not the whole explanation. The number of RCSs per 100,000 residents in the district varies from just under 11 to as many as 33 RCSs per 100,000 inhabitants, and on average, there are nearly 26 RCSs per 100,000 inhabitants in Stockholm.

If we instead analyse the number of RCSs per 1000 hectares of land area, it can be observed that the spread is significantly greater between the districts. The inner-city districts of Södermalm, Norrmalm and Kungsholmen all have a significantly higher RCS density

than those in the suburbs. The only exception is the inner-city district of Östermalm, which has relatively few RCSs, both measured as a proportion of the population and measured as land area density. The difference between Södermalm and Östermalm is surprisingly large.

**Table 1.** Number of recycling stations (RCSs) in relation to population.

| District | RCSs (2022) | RCSs per 100,000 Inhabitants | RCSs per 1000-Hectare Land Areal |
|---|---|---|---|
| Bromma | 18 | 22.2 | 7.3 |
| Enskede-Årsta-Vantör | 35 | 33.8 | 16.6 |
| Farsta | 17 | 28.4 | 11.0 |
| Hägersten-Liljeholmen-Älvsjö | 34 | 27.4 | 15.3 |
| Hässelby-Vällingby | 25 | 32.8 | 12.8 |
| Kungsholmen | 18 | 25.2 | 37.1 |
| Norrmalm | 16 | 21.7 | 32.5 |
| Rinkeby-Kista | 6 | 11.7 | 5.1 |
| Skarpnäck | 12 | 25.8 | 7.7 |
| Skärholmen | 11 | 29.7 | 12.4 |
| Spånga-Tensta | 8 | 20.8 | 6.2 |
| Södermalm | 34 | 25.9 | 42.5 |
| Östermalm | 16 | 20.2 | 8.9 |
| Total | 250 | 25.6 | 13.3 |

Source: FTI and the City of Stockholm. Own calculations.

Through FDI, we have received a list of addresses where the 250 RCSs in the municipality of Stockholm are located. All surveyed RCSs are small and are aimed at recycling in the local residential area; the space on the street or sidewalk around them is about $10 \times 3$ m. (see Figure 1). These addresses have been coordinated via Google Maps, and each location has been visually inspected through Google Street View. The pictures show the design of the recycling bins.

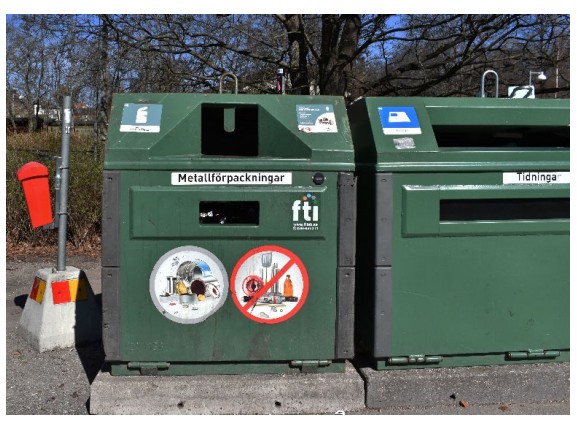
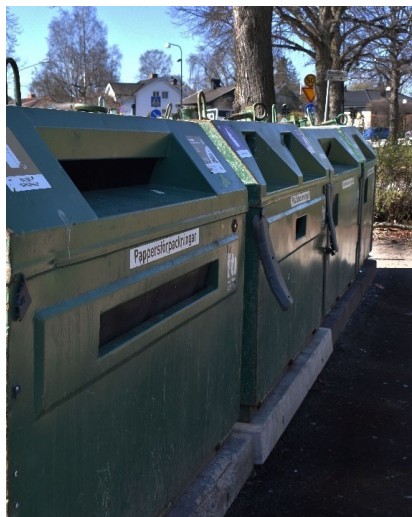

**Figure 1.** Pictures of recycling stations in the city of Stockholm. Photo: Emma Wilhelmsson.

All RCSs are located by a road to allow for the emptying of the containers. The majority of RCSs in the inner city are located on streets in residential areas, while outside the inner city, many stations are located on the exit road from residential areas. This means that many are located with other urban area features such as roads, petrol stations and subway stations, which are potential disamenities. However, several are also located near amenities such as parks and playgrounds. In Table 2, all RCSs are categorised depending on where they are located in terms of micro-location.

**Table 2.** Recycling stations and micro-location.

| Micro-Location | Number |
|---|---|
| Minor road | 167 |
| Major road | 22 |
| Gas station | 7 |
| Subway station | 16 |
| Shops, retail | 24 |
| Parking | 31 |
| Green area, sports facility, water | 67 |
| Playground, school | 28 |
| In a multi-family housing area | 143 |
| In a single-family housing area | 50 |
| Office, light industry | 23 |

Note. Each recycling station can be located in multiple micro-locations.

The majority of RCSs are located by a road to make it easier for residents to recycle, and facilitate the emptying and maintenance of the facilities. The streets are classified into three groups: a minor road means a smaller street with traffic that mainly consists of the area's residents; a road is a street with some pass-through traffic; and major roads are those with significant traffic volumes. RCSs are primarily located on minor roads, but several are located in car parks in residential areas. Relatively few are co-located with subway stations or local shops in the area. Only seven of the 250 RCSs are found at or near gas stations. It can also be noted that significantly more are located at parks and other green areas with playgrounds or directly adjacent to water. More RCSs are found in residential areas with multi-family houses than those with single-family houses. The conclusion that can be drawn is that RCSs are, above all, co-located with amenities rather than with disamenities.

### 3.3. Importance of RCS Location

Hage and Söderholm's (2008) [40] results indicate that proximity to RCS, population density, and the proportion of residents in urban areas have a small economic and statistical impact. Moreover, one recently published article by [10] shows that distance is not as crucial as might be expected. In an experiment in Shanghai, households had to register their interest in a recycling programme, and they did not find that the distance to the nearest recycling station had any effect on the households that chose to sign up for the programme. However, these results contradict [7,8,41]. Moreover, [42] show that increased density (RCS per capita) increases the degree of recycling in the municipality.

Hence, [7,8,41,42] all show that the distance to RCS is essential and that reduced distance significantly increases the sorting of recycled materials. Proximity to RCS is therefore crucial in city planning in both new and existing residential areas.

Moreover, there is a trade-off between how much space the city wants to cover, or population they want to reach, and the collection cost. Ref. [43] presents a mathematical model where the problem is to increase simultaneously the degree of recycling, while minimising installation and collection costs through the strategic location of RCSs. To this problem, the effect on housing attractiveness could be added. The authors of [12] analyse the optimal number and distribution of RCSs, and they also do not consider the negative externality of the stations in the form of deteriorating residential area attractiveness. They

only optimise recycled material as a function of population density and consumption to transport costs.

## 4. Data and Descriptive Statistics

Data regarding housing transactions come from Svensk Mäklarstatistik. A high proportion of brokers report contract data to them, who then compile and publish statistics on price development in Sweden and different geographical areas. The transactions this research project has access to are raw data included in all sales and data regarding owner-occupied cooperative apartments and single-family houses for 2005–2019. In total, we have access to over 200,000 transactions, and the majority of these are condominium sales.

Available data regarding condominiums are transaction price in SEK, contract date, the living area measured in square meters, number of rooms, monthly fee to the housing association in SEK, year of construction, and floor plan. Apart from the variables of monthly fee and floor plan, available data for single-family houses are the same, with the addition of information about plot area in square meres. For all transactions, we also have access to longitude and latitude; and distances in kilometres to the Central Business District (CBD), the nearest metro station, and the seventeen largest shopping centres in the Stockholm area [21]. The descriptive statistics are exhibited in Table 3.

**Table 3.** Descriptive statistics (condominium and the single-family sample).

| PANEL A | | | | | |
|---|---|---|---|---|---|
| *Variable* | *Obs* | *Mean* | *Std. Dev.* | *Min* | *Max* |
| *Price (SEK)* | 166,677 | 3,161,138.9 | 2,011,335 | 700,000 | 35,000,000 |
| *Living area (Square meter)* | 166,677 | 62.558 | 25.492 | 24.1 | 239 |
| *Monthly fee (SEK)* | 166,228 | 3192.649 | 1276.845 | 878 | 97,350 |
| *Number of rooms* | 166,677 | 2.371 | 0.993 | 1 | 7 |
| *Built (Year)* | 166,627 | 1952.248 | 33.36 | 1882 | 2019 |
| *Apartment floor level* | 166,677 | 2.561 | 1.743 | 0 | 9 |
| *Binary RCS* | 166,677 | 0.39 | 0.488 | 0 | 1 |
| *Dist RCS (kilometres)* | 164,817 | 0.29 | 1.967 | 0.001 | 674.782 |
| *Dist subway station (kilometres)* | 164,817 | 0.562 | 2 | 0.003 | 674.839 |
| *Dist shopping mall (kilometres)* | 164,817 | 1.459 | 2.132 | 0.006 | 663.341 |
| *Dist CBD (kilometres)* | 164,817 | 4.712 | 3.7 | 0.112 | 684.722 |
| PANEL B | | | | | |
| *Variable* | *Obs* | *Mean* | *Std. Dev.* | *Min* | *Max* |
| *Price (SEK)* | 15,213 | 4,858,830.1 | 2,322,568 | 700,000 | 20,800,000 |
| *Living area (square meters)* | 15,213 | 122.837 | 33.742 | 29 | 239 |
| *Plot area (square meters)* | 15,021 | 4992.447 | 544,369.09 | 76 | 66,718,464 |
| *Number of rooms* | 15,213 | 5.306 | 1.076 | 1 | 7 |
| *Built (Year)* | 15,211 | 1956.22 | 22.774 | 1888 | 2018 |
| *Binary RCS* | 15,213 | 0.156 | 0.363 | 0 | 1 |
| *Dist RCS (kilometres)* | 15,192 | 0.49 | 4.719 | 0.011 | 508.915 |
| *Dist subway station (kilometres)* | 15,192 | 1.322 | 4.766 | 0.021 | 509.009 |
| *Dist shopping mall (kilometres)* | 15,192 | 3.122 | 4.018 | 0.285 | 401.351 |
| *Dist CBD (kilometres)* | 15,192 | 10.104 | 5.698 | 1.005 | 525.572 |

Note. Table 3 shows the descriptive statistics concerning the two forms of housing transaction data used in the study: condominiums (Panel A) and single-family houses (Panel B). Shown in the table are the number of observations, mean value, standard deviation (Std. Dev.), minimum (Min) and maximum (Max) values. Price is measured in Swedish krona (SEK), living area and plot area in square metres and monthly fee in SEK. Built denotes the building year. RCS (recycling station) is a binary variable measuring the treatment that the transaction is within 200 m from the recycling station. Distance to RCS, subway station, shopping mall and Central Business District (CBD) is measured in kilometres. CBD in Stockholm is Sergel Torg. Potential outliers that have been excluded are observations with prices below the one percentile of the price distribution: the same concerns living area, monthly fee and building year. Condominiums with an apartment floor below 0 have been excluded.

In the analysis regarding condominiums, approximately 165,000 transactions will be used. Approximately 25,000 transactions are missing information about the apartment floor level, so fewer observations are used. Potential outliers have been excluded (see information in the note in Table 3). The number of transactions regarding single-family houses amounts to just over 15,000. The average price for single-family houses is almost

SEK 2 million higher, with an average price of approximately SEK 5 million. However, the standard deviation is high both in the case of condominiums and single-family houses. Nevertheless, the variation concerning the price is higher for condominiums, as there is a more significant variation in price ranges for condominiums than for single-family houses. The higher price is reflected in the fact that the size of single-family houses is almost twice as large as condominiums. Condominiums exist for natural reasons in the inner city, and thus the distance to the CBD is significantly higher for single-family houses than for condominiums. The same applies to proximity to metro stations and shopping centres. It is also clear that the distance to the nearest recycling station is significantly shorter for tenant-owner apartment transactions than for single-family housing transactions. For condominiums, the average distance to a recycling station is about 300 m compared to 500 m for single-family houses. However, the variation is significantly greater among single-family houses.

Proving the capitalisation effect on housing values from RCS exclusively will be challenging. Each recycling station has been related to housing transactions, and the shortest distance (measured as the bird flies in metres) to a recycling station has been registered for each transaction. This distance to RCS (*Dist RCS*) will be one of the area's attributes included in the hedonic price equation. The average distance equals 290 m for condominiums and 490 m for single-family houses. The variation around the average distance is considerable. A binary variable (*Binary RCS*) indicating whether the house is within 2 kilometres of the RCS has been included in the model. Nearly 40 percent of the condominiums are within 2 kilometres, but only 15 percent of the single-family houses.

Fixed effects have been included to minimise the risk of omitted variable bias. All models include fixed effects for time and postcode areas. The postcode areas are relatively small, and thus they are many-just over 700. Information concerning transactions with the same location, i.e., on the same property, will be utilised. In some cases, it is the same apartment, but in others, it involves apartments on the same property. Just under 50 percent of condominium transactions are transactions on the same property. Most of these relate to 2–4 transactions per property, but there are some with significantly more transactions. In fact, one property reports just over 300 transactions. It is a sizeable tenant-owner association, and the sales cover an extended period, so the number is not unreasonable. However, 95 percent of sales regard properties with 0–10 transactions. When it comes to single-family houses, sales of the same property are repeated sales. Significantly fewer have been sold more than once: as many as 78 percent have only been sold once; and about 22 percent have been sold 2–4 times. With the help of this information, fixed property effects have been created and included in the hedonic price equation. When it comes to the condominium apartment market, it has not been possible to include 100,000 fixed property effects. Therefore, only one model regarding the inner city has been estimated.

**5. Empirical Analysis**

The estimation of the hedonic price equation has been made in four different steps. In the first step, default models regarding condominiums and single-family houses have been estimated. The models include all residential types and area attributes previously presented. In addition to these, fixed time and area effects are also included, and the latter refers to postcode areas. Proximity to RCS is included partly as a continuous variable that measures the distance between the recycling station and the residential dwelling and partly as a binary variable in a simple treatment model where the variable is equal to 1 within 200 m, and the control area consists of 200 to 500 m from the recycling station. We have used the propensity score method to control for non-randomness. Results are shown in Table 4.

**Table 4.** Empirical results—Default models (OLS and WLS estimates).

|  | (1) | (2) | (3) | (4) |
|---|---|---|---|---|
|  | *Condominium* | *Condominium* | *Single-Family* | *Single-Family* |
| *Living area* | 0.0113 *** | 0.0102 *** | 0.00334 *** | 0.00263 *** |
|  | (288.86) | (160.98) | (47.15) | (33.93) |
| *Plot area* |  |  | $3.89 \times 10^{-10}$ | 0.000292 *** |
|  |  |  | (0.16) | (33.81) |
| *Monthly fee* | −0.0000298 *** | −0.00000166 *** |  |  |
|  | (−52.53) | (−12.92) |  |  |
| *Number of rooms* | 0.0583 *** | 0.0445 *** | 0.0460 *** | 0.0391 *** |
|  | (65.78) | (32.20) | (24.09) | (19.60) |
| *Built* | −0.000244 *** | −0.000115 *** | −0.00147 *** | 0.000137 |
|  | (−11.93) | (−4.33) | (−14.65) | (1.05) |
| *Apartment floor* | 0.0178 *** | 0.0162 *** |  |  |
|  | (76.42) | (54.48) |  |  |
| *Dist RCS* | 0.0735 *** |  | 0.0345 * |  |
|  | (14.16) |  | (2.20) |  |
| *Binary RCS* |  | −0.0137 *** |  | −0.0143 *** |
|  |  | (−11.33) |  | (−3.67) |
| *Dist subway station* | −0.0137 *** | −0.00298 | 0.0549 *** | 0.0419 *** |
|  | (−3.40) | (−0.48) | (5.25) | (3.97) |
| *Dist shopping mall* | 0.0418 *** | 0.0558 *** | 0.0545 *** | 0.0665 *** |
|  | (11.50) | (10.46) | (5.52) | (6.70) |
| *Dist CBD* | −0.0763 *** | −0.0808 *** | −0.0368 *** | −0.0258 ** |
|  | (−23.04) | (−15.23) | (−4.93) | (−3.21) |
| *Constant* | 14.17 *** | 14.00 *** | 17.37 *** | 14.35 *** |
|  | (295.78) | (238.63) | (69.50) | (56.69) |
| *Observations* | 144,902 | 144,902 | 10,594 | 10,593 |
| $R^2$ | 0.935 | 0.959 | 0.860 | 0.879 |
| *AIC* | −156,180.0 | −327,170.7 | −8855.8 | −0328.4 |

Note. Table 4 shows the ordinary least square estimates (OLS) concerning models 1 and 3 and weighted least square estimates (WLS) concerning models 2 and 4. The weights are based on the propensity score estimates belonging to the treatment group. Models 1 and 2 are the condominium apartment market, while models 3 and 4 address the single-family housing market. All models include fixed postal code effects and fixed monthly effects. Only observations within 500 m from the RCS are included in the estimations. The treatment (RCS) group are observations within 200 m of RCS, and the control group observations within 200 to 500 m. $t$ statistics in parentheses, * $p < 0.05$, ** $p < 0.01$, *** $p < 0.001$.

In step two, the hypothesis that the implicit price for a recycling station is affected by micro-location has been tested. The RCSs are classified into two groups based on the micro-location characteristics. The first refers to locations with disamenities (such as major roads or petrol stations) in their vicinity, and the second refers to whether they are located where there are amenities (such as parks). The results are shown in Table 5. In step three, we test if the estimates are affected by including fixed property effects, and those results are exhibited in Table 6. Finally, in step four, we are testing our assumption about treatment and control areas, as well as performing a placebo test concerning the location of RCSs. The results for this are presented in Tables A1 and A2 in the Appendix A.

**Table 5.** Empirical results—Micro location models.

|  | (1) | (2) |
|---|---|---|
|  | *Condominium* | *Single-Family* |
| *Living area* | 0.0102 *** | 0.00263 *** |
|  | (161.04) | (33.93) |
| *Plot area* |  | 0.000292 *** |
|  |  | (33.76) |
| *Monthly fee* | −0.00000165 *** |  |
|  | (−12.77) |  |

**Table 5.** *Cont.*

|  | **(1)** | **(2)** |
|---|---|---|
|  | *Condominium* | *Single-Family* |
| Number of rooms | 0.0445 *** | 0.0390 *** |
|  | (32.22) | (19.60) |
| Built | −0.000119 *** | 0.000134 |
|  | (−4.48) | (1.03) |
| Apartment floor | 0.0162 *** |  |
|  | (54.45) |  |
| Binary RCS | −0.00679 *** | −0.0182 ** |
|  | (−4.52) | (−3.20) |
| Binary RCS (good) | −0.0102 *** | 0.0125 |
|  | (−4.36) | (1.39) |
| Binary RCS (bad) | −0.0160 *** | 0.00288 |
|  | (−6.73) | (0.32) |
| Dist subway station | −0.00401 | 0.0423 *** |
|  | (−0.65) | (3.99) |
| Dist shopping mall | 0.0570 *** | 0.0668 *** |
|  | (10.71) | (6.72) |
| Dist CBD | −0.0797 *** | −0.0262 ** |
|  | (−15.07) | (−3.25) |
| Constant | 14.00 *** | 14.35 *** |
|  | (238.77) | (56.65) |
| Observations | 144,902 | 10,593 |
| $R^2$ | 0.959 | 0.879 |
| AIC | −327,232.6 | −10,328.5 |

Note. Table 5 shows the weighted least square estimates (WLS) concerning models that address both the (1) condominium and (2) single-family housing market. The weights are based on the propensity score estimates belonging to the treatment group. All models included fixed postal code effects and fixed monthly effects. Only observations within 500 m of the RCS are included in the estimations. The treatment (RCS) group are observations within 200 m of RCS, and the control group observations within 200 to 500 m. RCS is a binary variable measuring whether the observation is close to any RCS, an RCS with amenities (good), or and RCS with disamenities (bad). $t$ statistics are in parentheses, * $p < 0.05$, ** $p < 0.01$, *** $p < 0.001$.

**Table 6.** Empirical results—Property fixed effects models.

|  | **(1)** | **(2)** |
|---|---|---|
|  | *Condominium* | *Single-Family* |
| Living area | 0.0101 *** | 0.00262 *** |
|  | (50.94) | (29.50) |
| Monthly fee | −0.00000306 |  |
|  | (−0.99) |  |
| Number of rooms | 0.0878 *** | 0.0382 *** |
|  | (26.36) | (16.95) |
| Built | −0.000609 *** | 0.000183 |
|  | (−8.26) | (1.25) |
| Apartment floor | 0.0201 *** |  |
|  | (28.72) |  |
| Binary RCS | −0.0316 *** | −0.0126 ** |
|  | (−8.34) | (−2.70) |
| Dist subway station | 0.0526 * | 0.0556 *** |
|  | (2.45) | (4.48) |
| Dist shopping mall | 0.0975 *** | 0.0590 *** |
|  | (5.57) | (5.10) |
| Dist CBD | −0.00947 | −0.0325 *** |
|  | (−0.79) | (−3.51) |
| Constant | 14.80 *** | 14.28 *** |
|  | (102.30) | (50.07) |

**Table 6.** *Cont.*

|  | (1) | (2) |
|---|---|---|
|  | *Condominium* | *Single-Family* |
| *Observations* | 22,296 | 10,593 |
| $R^2$ | 0.950 | 0.902 |
| *AIC* | −27,296.3 | −10,643.5 |

Note. Table 6 shows the condominium (1) and single-family (2) housing market's weighted least square estimates (WLS). The weights are based on the propensity score estimates belonging to the treatment group. All models include fixed postal code effects, fixed monthly effects, and fixed property effects. The condominium sample can be repeated sales or other sales in the same building; and in the single-family sample, it is repeated sales. Only observations within 500 m of RCS are included in the estimations. The treatment (RCS) group are observations within 200 m of RCS, and the control group observations within 200 to 500 m. *t* statistics are in parentheses, * $p < 0.05$, ** $p < 0.01$, *** $p < 0.001$.

### 5.1. Default Models

The estimated default models are presented in Table 4. The owner-occupied condominium market is analysed in the first two columns, and in the last two columns, the results from the single-family housing market are presented. Proximity to RCS has been estimated in two different ways, namely (1) as a continuous variable (*Dist RCS*) and (2) as a binary treatment variable (*Binary RCS*).

All models include fixed postal codes and monthly effects, and the degree of explanation is generally high. The explanation rate in the condominium models is approximately 95 percent, while the explanation rate is approximately 87 percent for the single-family housing market. The degree of explanation is slightly higher in the treatment effect models, where the proximity to RCS is a binary variable. The model is weighted based on the probability that the transaction has a treatment. All estimates have an expected effect on prices and reasonable magnitude. Both increased living area in square metres and number of rooms raise prices. Older houses have a lower expected price, and for condominiums, a higher monthly fee to the housing association has a negative impact on the price. Proximity to the metro station positively impacts apartment prices but has a negative impact on single-family housing prices. Proximity to the CBD positively affects prices, while proximity to the shopping mall does not.

The variable of interest here is proximity to RCS. All estimates are statistically significant at a 5% significance level with *t*-values of around 10 (absolute values) in the condominium model but slightly lower in the single-family housing model (around 2–3 in final values). Proximity also has an expected effect to the extent that it can be regarded as a disamenity. The closer to the RCS the dwelling is located, the lower the price, everything else being equal. In the binary models, the effect amounts to approximately 1.3 percent of the housing values, which can also be regarded as economically significant. Measured in SEK, the capitalisation amounts to approximately 40,000 for condominiums and 70,000 for single-family houses (around 3800 and 6800 EUR). Our results are in line with [15] but significantly lower than, e.g., [13,14].

### 5.2. Micro-Location Models

To test the significance of the micro-location, two interaction variables in the model where RCS has been integrated with amenities and disamenities have been included. Amenities mean that RCSs are located close to a park or other green area, while disamenities mean that RCSs are co-located near, for example, major roads, gas stations and retail trade. The results from the estimates are shown in Table 5.

Results indicate that the co-location of RCSs with amenities and disamenities has some impact on capitalisation. Both the condominium (column 1) and single-family (column 2) housing analyses have been estimated with weighted least square, where the weights consist of the probability of being included in the treated group. The degree of explanation is high, as it is in the default model. Around 95 percent of the variation in condominium

prices can be explained by the included variables, as well as around 88 percent of the variation in single-family house prices.

Proximity to RCS has a negative impact on prices. The effect is more evident in the single-family house segment than in the condominium one. In the single-family housing market, we can also note that the negative capitalisation effect is independent of whether RCS is co-located with other disamenities or, for that matter, with amenities. This is not the case in the condominium segment, where there is a negative capitalisation regarding proximity to RCS. However, it is higher if RCSs are co-located with positive characteristics in the residential environment, such as proximity to parks and other green areas. Moreover, it is also significantly higher negatively capitalised if RCSs are co-located with negative characteristics in the residential environment, such as proximity to major roads or gas stations. Our results indicate that RCSs have a negative impact on condominium prices, but this is especially clear if there are other disamenities in their vicinity. Thus, one could conclude that in an urban environment, the effect of RCSs is relatively limited if these have not been placed in environments that are otherwise considered attractive. In the single-family housing market, the proximity to an RCS is negatively capitalised into housing values regardless of its micro-location.

### 5.3. Property Fixed Effects

Additional fixed effects at the property level to reduce the risk of omitted variables and endogeneity problems have been added. For the owner-occupied apartment market, we have added fixed effects for the building, which in most cases refers to neighbouring condominiums in the same properties but also repeated sales. For the single-family housing market, we have added fixed property effects that, in all cases, refer to repeated sales. The results are presented in Table 6.

The number of independent variables increases dramatically from just over 200 in the model with fixed postal codes and monthly effects, to over 700 in the model with fixed property effects. The degree of explanation in the model increases slightly, but it is not a statistically significant difference. The capitalisation of proximity to RCS increases in the inner city, where more condominiums are located, while the effect in the single-family areas is equivalent to the model without the property fixed effects. An RCS within 200 m doubles from 1.3 percent to 3.2 percent in the condominium market. The inclusion of the fixed property effects thus has a substantial effect on capitalisation. It can also be stated that the implicit prices regarding proximity to the subway station and CBD change dramatically in the model that explains the condominium prices, but not in the single-family house model. The fixed property effects effectively pick up the effect of proximity to the CBD (no longer statistically significant) and subway station (statistically significant but with a changed sign).

### 5.4. Robustness Test

It is assumed that the effect of RCS is local to the extent that we have assumed that the treatment range is from 0 to 200 m, and the control group consists of 200 to 500 m. To test how robust these assumptions are, alternative intervals, namely 0–100 m and 0–300 m as a treatment group and 100–400 m and 300–600 m as a control group, respectively, have been tested. The results are shown in Table A1 in the Appendix A.

The estimates are robust concerning assumptions about the size of the treatment and control group. A narrower treatment group results in equivalent estimates in the condominium analysis, even if the statistical significance is somewhat greater in the narrower range than in the broader one. The estimates in the single-family housing sample are slightly larger in the broader range than in the narrower one, but the difference is not statistically significant. However, the robustness test indicates that the capitalisation effect is not dependent on the assumption of treatment and the size of the control group. The degree of explanation is somewhat higher in the narrower range, but the differences are minimal in the condominium sample and slightly larger in the single-family house sample.

As a further robustness test, we have randomised where the 250 RCSs are located. The RCSs have been randomly "moved" 0–500 m from their actual location. The distance between the housing transactions and the "new" location is then re-calculated. The methodology is inspired by the so-called placebo test used in the regression discontinuity design methodology. The hypothesis is that this random movement of RCS does not have a negative capitalisation on prices.

The results from these analyses can be found in Table A2 in the Appendix A and show that the effect of RCS is now statistically insignificant. This strengthens our interpretation that the proximity of RCS has a statistically significant negative causal effect on housing value, i.e., no placebo effect, which means that RCSs harms housing attractiveness.

## 6. Conclusions and Policy Implications

This paper discusses the importance of planning where RCSs should be located in the urban context. In our analysis of whether the location of RCSs has a negative effect on the residential environment, a traditional hedonic methodology is used where the value of the dwelling is built up by the housing attributes. The most important of these are attributes and characteristics associated with the house and its location in the city. However, attributes in the residential environment also impact the value, including attributes such as proximity to green areas and various private and public services. There are also many negative externalities in the city that can affect the attractiveness of the residential area. Proximity to RCS is one of these disamenities as they can cause disturbances in litter, noise, odour, and pollution.

Almost 200,000 housing transactions in Stockholm, Sweden, are utilised to estimate the hedonic price equation. Great care has been taken to ensure that the estimated relationships are also causal relationships by including different types of fixed effects, analysing the environment where the RCSs are located and including as many value-adding attributes as possible in the model. This has been done to minimise omitted variable bias, control for reverse causality and selection bias in treatment and minimise the extent of measurement error.

The negative capitalisation of RCS in property values shows that it is an important parameter when planning how many and where RCSs are to be placed in the urban environment. The co-location with other disruptive activities seems obvious, but this is not always the case today. It also indicates that the design of the RCS and their management are components that may affect capitalisation and, therefore, essential to consider.

What policy implications come from the results of this analysis? Previous analyses show that the location impacts the cost of both emptying and maintaining the RCS, there being a trade-off which impacts the frequency with which households will use the stations. The fewer the RCSs that are set up in the city, the lower the operating cost, but at the same time, this reduces recycling. Our empirical analysis shows that RCSs impact the attractiveness of the residential area. Therefore, optimising where and how many RCSs are put up in the city should include these negative social costs in the objective function.

**Funding:** We would like to thank the research project Housing 2.0 (Bostad 2.0) for financial support.

**Acknowledgments:** We would like to thank Svensk Mäklarstatistik AB for transaction data.

**Conflicts of Interest:** The authors declare no conflict of interest.

## Appendix A

**Table A1.** Robustness test (treatment and control groups).

|  | **(1)** | **(2)** | **(3)** | **(4)** |
|---|---|---|---|---|
|  | **100 Meters** | **300 Meters** | **100 Meters** | **300 Meters** |
| Living area | 0.0102 *** | 0.0101 *** | 0.00265 *** | 0.00313 *** |
|  | (152.61) | (165.43) | (29.07) | (41.38) |

**Table A1.** *Cont.*

| | (1) | (2) | (3) | (4) |
|---|---|---|---|---|
| | 100 Meters | 300 Meters | 100 Meters | 300 Meters |
| Plot area | | | 0.000298 *** | −0.00000413 *** |
| | | | (30.37) | (−16.13) |
| Monthly fee | −0.00000174 *** | −0.00000180 *** | | |
| | (−9.59) | (−13.53) | | |
| Number of rooms | 0.0463 *** | 0.0444 *** | 0.0357 *** | 0.0458 *** |
| | (32.00) | (33.18) | (15.87) | (23.42) |
| Built | −0.000187 *** | −0.0000876 *** | 0.000238 | −0.00141 *** |
| | (−6.81) | (−3.37) | (1.61) | (−12.20) |
| Apartment floor | 0.0163 *** | 0.0165 *** | | |
| | (51.87) | (57.75) | | |
| Binary RCS | −0.0134 *** | −0.0138 *** | −0.0155 *** | −0.0172 *** |
| | (−10.98) | (−11.43) | (−3.90) | (−4.09) |
| Dist subway station | −0.0319 *** | 0.00718 | 0.0206 | 0.0608 *** |
| | (−4.33) | (1.22) | (1.64) | (6.11) |
| Dist shopping mall | 0.0526 *** | 0.0489 *** | 0.0655 *** | 0.0551 *** |
| | (8.90) | (9.50) | (5.26) | (5.84) |
| Dist CBD | −0.0828 *** | −0.0831 *** | −0.0328 *** | −0.0308 *** |
| | (−13.42) | (−16.06) | (−3.52) | (−4.02) |
| Constant | 14.13 *** | 13.94 *** | 14.20 *** | 17.59 *** |
| | (236.67) | (241.42) | (49.73) | (41.00) |
| Observations | 131,277 | 153,157 | 7978 | 12,479 |
| $R^2$ | 0.960 | 0.959 | 0.884 | 0.852 |
| AIC | −308,730.1 | −338,308.5 | −8047.8 | −299,995.4 |

Note. Table A1 shows the condominium (1 and 2) and single-family (3 and 4) housing market's weighted least square estimates (WLS). The weights are based on the propensity score estimates belonging to the treatment group. All models include fixed postal code effects and fixed monthly effects. Only observations within 4–600 meters of the RCS are included in the estimations. The treatment (RCS) groups are observations within 100 and 200 meters of RCS, and the control group observations within 100 to 600 meters. $t$ statistics are in parentheses, * $p < 0.05$, ** $p < 0.01$, *** $p < 0.001$.

**Table A2.** Robustness test (placebo effect).

| | (1) | (2) |
|---|---|---|
| | Condominiums | Single−Family Houses |
| Living area | 0.0113 *** | 0.00335 *** |
| | (288.77) | (47.25) |
| Monthly fee | −0.0000299 *** | |
| | (−52.64) | |
| Number of rooms | 0.0583 *** | 0.0460 *** |
| | (65.80) | (24.10) |
| Built | −0.000231 *** | −0.00147 *** |
| | (−11.26) | (−14.65) |
| Apartment floor | 0.0177 *** | |
| | (76.00) | |
| Binary Placebo RCS | 0.00648 | 0.0121 |
| | (1.21) | (0.62) |
| Dist subway station | −0.0119 ** | 0.0542 *** |
| | (−2.96) | (5.17) |
| Dist shopping mall | 0.0410 *** | 0.0545 *** |
| | (11.26) | (5.52) |
| Dist CBD | −0.0755 *** | −0.0370 *** |
| | (−22.79) | (−4.95) |
| Plot area | | $4.44 \times 10^{-10}$ |
| | | (0.19) |
| Constant | 14.16 *** | 17.38 *** |
| | (295.56) | (69.51) |

**Table A2.** *Cont.*

| | (1) | (2) |
|---|---|---|
| | **Condominiums** | **Single−Family Houses** |
| Observations | 144,902 | 10,594 |
| $R^2$ | 0.935 | 0.860 |
| AIC | −155,980.0 | −8851.2 |

Note. Table A2 shows the condominium (1 and 2) and single-family (3 and 4) housing market's weighted least square estimates (WLS). The weights are based on the propensity score estimates belonging to the treatment group. The variable binary Placebo RCS is based on the distance from the dwelling to the recycle stations randomly moved from 0-500 meters. All models include fixed postal code effects and fixed monthly effects. *t* statistics are in parentheses * $p < 0.05$, ** $p < 0.01$, *** $p < 0.001$.

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
