# Peer review of "About the Importance of Planning the Location of Recycling Stations in the Urban Context"

_sustainability, doi:10.3390/su14137613_

Round 1
Reviewer 1 Report
Comments on “About the importance of planning the location of recycling stations in the urban context”.
Dear Authors,
The paper must be significantly improved. Please consider the following remarks:
Major comments:
(1) Please explain novelty. Please compare results with other studies. You can use table in Introduction part.
(2) Please improve abstract part. Please add specific results. What does “affect housing prices” mean?
(3) Introduction part: Literature review should be do more deeply.
(4) Please add discussion part
(5) Line 298 – 299. “The difference between Södermalm and Östermalm is surprisingly significant.” Please explain.
(6) Please explain “We have used almost 200,000 housing transactions in Stockholm, Sweden, to estimate the hedonic price equation. We have taken great care to ensure that the relationships we have estimated are also causal relationships by including different types of fixed effects, analysing the environment where the RCS are located and including as many value-adding attributes as possible in the model. This has been done to minimise omitted variable bias, control for reverse” Please explain. It is only case study
Minor comments:
(1) Line 37, 157: Please avoid using lumped references. The references must be cited one by one showing what is new in the present publication with respect to the cited reference
(2) Line 157: Please improve the way of calling references. Please see journal template. The same approach I suggest applying also in the rest of the manuscript.
(3) Line 16, 164: First person pronouns and verbs suggest that authors may be too close to the subject matter and is mixing opinion with fact or may even be hiding something. Please reduce the use of the first person (we) in academic writing. The same approach I suggest applying also in the rest of the manuscript.
(4) Please add nomenclature and abbreviations table
(5) Keywords. Please expand list / please improve - Authors should make better use of this section to allow the article to be found on search engines. Please add “Sweden”, Stockholm, Case Study
(6) Map 1. In my opinion is not relevant. Please add legend, scale, some specific data (for example RCS location, area, number of inhabitants.
(7) Table 1. You can add area, number of inhabitants
(8) Table 2. “patrol station” Please confirm
(9) Table 3. Please improve the title. Please add units for variable
(10) Table 4, 5, 6. Please explain red color
(11) Please improve reference part in line with journal template.
Author Response
Reviewer 1
Thank you for all very good comments. I have done most of the changes you require.
(1) One of the contributions of the empirical analysis is that there are not as many studies that analyse the effect of recycling stations on attractiveness. Many studies exist on the importance of accessibility, easy access and visibility for how much is recycled. However, not as much about the cost of residential area attractiveness. We have tried to emphasise that a little more in the text.
(2) we have added that" the effect amounts to approximately 1.3 percent of the housing values"
(3) As said earlier, there are few papers in this research area. However, we added a few more recent articles in the introduction. We have added:
"Miller et al. (2016) analyze the effect of accessibility to recycling stations and its visibility on how much is not recycled. As a case study, they use recycling in university buildings. Their results indicate that neither availability nor visibility significantly impacts recycling. The results, however, indicate that the combination of accessibility and visibility is important. Li et al.'s (2020) show similar results. DiGiacomo et al.'s (2017) research, on the other hand, shows that accessibility and easy access to recycling has a major impact on the behaviour of households in faulty family homes. They conclude that the result impacts waste management and environmental policy.
The design of the recycling stations not only has a potential negative impact on how much is recycled. Keramitsoglou and Tsagaraki's (2018) research shows the potential of involving the residents in designing the recycling stations as it both increases the acceptance of the recycling stations in the living environment and potentially increases the recycling. The design is also something that Jiang et al. (2021) point out in their research."
(4) Discussion part is included in the empirical analysis
(5) we have changed it to large. "The difference between Södermalm and Östermalm is surprisingly large."
(6) Not sure if I understand the question. We have a long section about the importance of controlling for omitted variable bias, measurement errors and reverse causality in the paper (6 pages). Do I need to write more? What do you mean by "it is just a case study"? I believe that regardless if it is one of many case studies, the importance of endogeneity in the models is vital.
Minor comments
- Sometimes lump references can be justified. This is such a case.
- References have been changed
- Deleted a large number of we. I did not know that I write we so many times.
- Not sure what is meant. All abbreviations are clearly defined in the text and the notes under the tables.
- Sweden and Stockholm, and circular economy are added.
- Has been deleted
- Adding area and inhabitants is not essential for the analysis and has not been added.
- Changed to gas station
- It has been added, but everything is in the note to the table.
- Changed to black.
- Done
Reviewer 2 Report
The content is in line with the mission and objectives of the target journal by dealing with circular economy and in particular recycling as well as the attractiveness of local areas. The article argues that the location of recycling stations is important for creating opportunities for the efficient collection and recycling of packaging materials.
Below I provide a few indications to improve the contribution in line with the high standards of the journal:
- keywords could be added. For instance "circular economy" could be added to the list proposed.
- the opening of paragraph 2. "theoretical and methodical framework" could benefit from an explanatory opening of the context of the study, briefly recalling the topic addressed.
- the literature review lacks key articles addressing the topic which were published in the target journal. Some examples that could be mentioned in the introduction or literature review are suggested below:
Shaban, Ahmed, Fatma-Elzahraa Zaki, Islam H. Afefy, Giulio Di Gravio, Andrea Falegnami, and Riccardo Patriarca. 2022. "An Optimization Model for the Design of a Sustainable Municipal Solid Waste Management System" Sustainability 14, no. 10: 6345. https://doi.org/10.3390/su14106345
Sastre, R.M.; de Paula, I.C.; Echeveste, M.E.S. A Systematic Literature Review on Packaging Sustainability: Contents, Opportunities, and Guidelines. Sustainability 2022, 14, 6727. https://doi.org/10.3390/su14116727
Structure:
check if they are some mistakes in the structure of sub-paragraphs on pages 15-16. It is indicated (a) property fixed effects and again a) robustness test.
In general, it should be pointed out better - even just indicating it as a title - the part dealing with the findings of the article and providing a clear heading of just a few sentences on how they will be presented. It can be challenging to follow the presentation of the empirical results.
The discussion paragraph seems to be missing and integrated with the presentation of the findings. It would be useful following also the structure of the articles published in Sustainability to include a specific paragraph where results are comprehensively discussed and also recall sources from the literature review mentioned in the introduction and theoretical framework. This paragraph depending on its length could be integrated with the conclusions if more appropriate.
The quality of language is high. There are some minor typos to check for example (p.1 will be need to be more).
- future research perspectives and limitations of the study should be addressed in the conclusions.
-
Author Response
Reviewer 2.
Thank you for your comments and suggestion of literature to include in the introduction.
---
Below I provide a few indications to improve the contribution in line with the high standards of the journal:
- keywords could be added. For instance "circular economy" could be added to the list proposed.
Answer: has been done
- the opening of paragraph 2. "theoretical and methodical framework" could benefit from an explanatory opening of the context of the study, briefly recalling the topic addressed.
Answer: Added “The research aims to increase knowledge about the recycling centres' local impact on attractiveness. We do this by analysing the prices of homes with ownership.”
- the literature review lacks key articles addressing the topic which were published in the target journal. Some examples that could be mentioned in the introduction or literature review are suggested below:
Shaban, Ahmed, Fatma-Elzahraa Zaki, Islam H. Afefy, Giulio Di Gravio, Andrea Falegnami, and Riccardo Patriarca. 2022. "An Optimization Model for the Design of a Sustainable Municipal Solid Waste Management System" Sustainability 14, no. 10: 6345. https://doi.org/10.3390/su14106345
Sastre, R.M.; de Paula, I.C.; Echeveste, M.E.S. A Systematic Literature Review on Packaging Sustainability: Contents, Opportunities, and Guidelines. Sustainability 2022, 14, 6727. https://doi.org/10.3390/su14116727
Answer: Both has been added.
Structure:
check if they are some mistakes in the structure of sub-paragraphs on pages 15-16. It is indicated (a) property fixed effects and again a) robustness test.
Avswer: checked.
In general, it should be pointed out better - even just indicating it as a title - the part dealing with the findings of the article and providing a clear heading of just a few sentences on how they will be presented. It can be challenging to follow the presentation of the empirical results.
Answer: not sure what you mean, but I have added a sentence in the abstract about the result. “(the effect amounts to approximately 1.3 percent of the housing values)”
The discussion paragraph seems to be missing and integrated with the presentation of the findings. It would be useful following also the structure of the articles published in Sustainability to include a specific paragraph where results are comprehensively discussed and also recall sources from the literature review mentioned in the introduction and theoretical framework. This paragraph depending on its length could be integrated with the conclusions if more appropriate.
Answer: Yes, I know the structure, e.g. Sustainability is different. I usually integrate my discussion in the result and concluding section, and I have not made any changes to the paper.
The quality of language is high. There are some minor typos to check for example (p.1 will be need to be more).
Answer: One extra read through has been done.
Reviewer 3 Report
The article is very appropriate as a part of the special issue "Circular Economy Approaches for Lifecycles of Products and Services".
It represents a very good case study on the issue of quantification, localization, and capitalization of recycling stations for urban waste. Proposals of complex solutions to the future are very beneficial because they can represent the result of the utilization of more exact models and descriptive statistics.
The research sample is representative and correctly structured, in relation to relevancy of conclusions from the research and realizations of proposals.
From the article, it is clear that this research is long-time and continual and in this research, the author tries to find new models for sustainability in the circular economy.
Planning, localization, and acceptability of recycling stations in the urban context have closely connected to be in the relation to the sustainable development also to the multicultural structure, and environmental awareness in the selected locality. The next research and subsequent activities of urban management can be oriented toward the capitalization of urban living.
Author Response
Thank you!
Round 2
Reviewer 1 Report
Accept
Reference part should be improved
Reviewer 2 Report
The Author has revised the article accordingly.